# Current and Future Therapeutical Options in Alport Syndrome

**DOI:** 10.3390/ijms24065522

**Published:** 2023-03-14

**Authors:** Jana Reiterová, Vladimír Tesař

**Affiliations:** 1Department of Nephrology, First Faculty of Medicine, Charles University, General University Hospital in Prague, 128 08 Prague, Czech Republic; 2First Faculty of Medicine, Institute of Biology and Medical Genetics, Charles University, General University Hospital in Prague, 128 08 Prague, Czech Republic

**Keywords:** Alport syndrome, therapy, gene

## Abstract

Alport syndrome (AS) is a hereditary kidney disease caused by pathogenic variants in *COL4A3* and *COL4A4* genes with autosomal recessive or autosomal dominant transmission or in the *COL4A5* gene with X-linked inheritance. Digenic inheritance was also described. Clinically it is associated with microscopic hematuria, followed by proteinuria and chronic renal insufficiency with end-stage renal disease in young adults. Nowadays, there is no curative treatment available. The inhibitors of RAS (renin-angiotensin system) since childhood slow the progression of the disease. Sodium-glucose cotransporter-2 inhibitors seem to be promising drugs from DAPA-CKD (dapagliflozin–chronic kidney disease) study, but only a limited number of patients with Alport syndrome was included. Endothelin type A receptor and angiotensin II type 1 receptor combined inhibitors, and lipid-lowering agents are used in ongoing studies in patients with AS and focal segmental glomerulosclerosis (FSGS). Hydroxychloroquine in AS is studied in a clinical trial in China. Molecular genetic diagnosis of AS is crucial not only for prognosis prediction but also for future therapeutic options. Different types of mutations will require various types of gene, RNA, or protein therapy to improve the function, the of final protein product.

## 1. Introduction

Alport syndrome (AS) is a hereditary kidney disease affecting glomeruli that leads to proteinuria followed by kidney failure [1]. Structural abnormalities of the glomerular basement membrane (GBM) are caused by mutations in *COL4A3* and *COL4A4* genes on chromosome 2 and *COL4A5* gene on chromosome X. These mutations result in the production of defective IV collagen α3, α4, or α5 chains, respectively, which is associated with the inappropriate assembly of the collagen chains in GBM. During the development of the kidney, the combination of the α1-α1-α2 network is present, later; in mature kidney, this network is replaced by α3-α4-α5 in GBM and by α5-α6-α5 in Bowman’s capsule [2]. The embryological components (α1-α1-α2) stay persistent in Alport syndrome due to the pathogenic variants and lead to a less functional GBM. The pathogenic variants in any of the three *COL4* genes are associated with vulnerability to filtration pressure and finally lead to irreversible damage of the glomerulus.

## 2. Prevalence and Genetic Background

The prevalence of Alport syndrome is estimated to be from 1 in 5000 to 1–9 in 100,000, according to ORPHANET. The true prevalence is probably higher. Predicted pathogenic variants of the *COL4A5* gene were established in one out of 2320 individuals, and *COL4A3* and *COL4A4* variants in one in 106 individuals in populations without kidney disease [3]. A recent study evaluating cases of unknown chronic renal diseases found monogenic nephropathies in 21% of cases, with one-third of hereditary glomerulopathies, which are most frequently caused by *COL4* genes pathogenic variants [4].

X-linked Alport syndrome (XLAS) caused by pathogenic variants in *COL4A5* gen accounts for less than 80% of patients with Alport syndrome. De novo *COL4A5* pathogenic variants develop in 10–15% of cases with negative family history. Despite the fact that heterozygous females usually have usually less severe clinical courses [5], there are nowadays recommended to be classified as having Alport syndrome instead of being carriers of Alport syndrome. Autosomal recessive Alport syndrome (ARAS) is caused by homozygous or compound heterozygous mutations in *COL4A3* and *COL4A4* genes. ARAS accounts for 5% of all cases of Alport syndrome. The most controversial is autosomal dominant Alport syndrome (ADAS) in patients with heterozygous mutation in *COL4A4* or *COL4A3* gene. The number of ADAS patients with end-stage renal disease in older age is increasing. The diagnosis of thin membrane nephropathy was recommended to be abandoned [6]. Patients with pathogenic variants in more than one *COL4* gene (digenic or trigenic inheritance), e.g., patients with digenic *COL4A4* and *COL4A3* mutations on different chromosomes (trans position) or on the same chromosome (cis position), demonstrate a clinical course similar to ARAS (trans position) or as ADAS (cis position) [7]. More studies evaluating the prognosis of patients’ digenic or trigenic mutations are necessary. The current classification of AS and risk to end-stage renal disease (ESRD) is summarized in Table 1.

Pathogenic variants in *COL4A4* or *COL4A3* genes were frequently found in patients with focal segmental glomerulosclerosis (FSGS). The *COL4* mutations were identified in 38% of patients with FSGS and positive family history [8]. Patients with FSGS and mutation in COL4A3, A4, or A5 should be classified as patients with Alport syndrome, and we should not use any immunosuppressive therapy because of its apparent ineffectiveness. On the other hand, mutational analysis of COL4 genes is recommended in FSGS patients resistant to immunosuppressive therapy.

## 3. Clinical Course and Progression

AS contains a wide spectrum of phenotypes. Thinning and thickening with the lamellated appearance of GBM is a typical finding described on electron microscopy in the sample from a renal biopsy. The result from renal biopsy does not help us with a prognosis, the risk of AS progression, the type of mutation in XLAS men, but also on the mode of inheritance. Genotype-phenotype correlations are well established for XLAS men but not for others forms.

Males with XLAS and all patients with ARAS suffer from hematuria, followed by microalbuminuria and proteinuria since childhood; 90% of these patients progress to ESRD by the age of 40. Large deletions and nonsense mutations in the *COL4A5* gene are associated with the highest risk of progression to ESRD by the age of 25 [8]. XLAS in females leads to ESRD in 40% by the age of 80 [9]. Males and females with ARAS usually progress to ESRD by the age of 40 [10].

Patients with ADAS and heterozygous females with XLAS exhibit a wide range of clinical courses not only among families but as well inside a family with the same mutation. Heterozygous mutations in *COL4A4* or *COL4A3* genes are frequent among patients with microscopic hematuria of glomerular origin and have been confirmed in patients with focal segmental glomerulosclerosis. All patients with heterozygous mutations should be regularly checked by nephrologists because they can progress to ESRD at an older age. The estimated risk for ESRD in patients with ADAS and risk factors such as proteinuria, FSGS, lamellation of GBM in renal biopsy, and sensorineural hearing loss is more than 20% [6]. Younger patients with one mutation in *COL4A4, A3* genes or females with one *COL4A5* mutation with microalbuminuria or even proteinuria should also be treated with RAS inhibitors, and we should always think about recruitment into clinical trials.

## 4. Treatment

A heterogeneous clinical course needs the cooperation of nephrologists, pathologists, and clinical and molecular genetics to find out an accurate diagnosis than is mandatory for therapeutical strategies. A detailed family history is helpful from nephrologists; the diagnosis can be confirmed by electron microscopy after a renal biopsy by a pathologist. Genetic testing can identify a pathogenic gene variant leading to individualized treatment in the future. On the other hand, we have to take into account that the mutation is not even with new generation sequencing identified in about 10% of patients with AS.

Young patients (children and adults according to study entry criteria) with AS without severe comorbidities such as diabetes mellitus, cardiovascular disease, and tumors are eligible candidates for clinical trials. The ongoing clinical trials are summarized in Table 2. Any effective drug for AS would be approved fast as an orphan drug. AS is a progressive glomerular disease with proteinuria, so AS could be used as a model for other kidney diseases with proteinuria. The treatment possibilities on different genetic (DNA, RNA, protein) and renal levels are summarized in Figure 1.

## 5. RAAS (Renin-Angiotensin-Aldosterone System) Inhibition

RAAS inhibition is widely used in all renal diseases with proteinuria. A significant increase in the renal angiotensin II, renin, and angiotensinogen was described in the kidneys of *Col4α3* knock-out mice [11]. Angiotensin converting enzyme inhibitor (ACEI) ramipril was studied in a mouse model with ARAS. Untreated animals died at 10 weeks; ramipril decreased proteinuria and kidney fibrosis and significantly prolonged survival [12].

Later onset of ESRD in males with XLAS was found in a cohort of 174 patients treated with ACEI in comparison with 109 untreated patients during two decades [13]. AT1 receptor blocker losartan was also found to be effective and safe in children with AS and proteinuria in comparison with placebo or amlodipine [14]. The EARLY PROTECT trial affirmed ramipril as a safe and effective drug in children with AS older than 2 years with isolated microscopic haematuria or with concomitant microalbuminuria. Ramipril diminished the slope of albuminuria progression, the decline of glomerular filtration rate, and the risk of disease progression almost to 50%. The efficacy of ACEI based on data from the European Alport Registry was also found in heterozygous AS carriers. The onset of ESRD occurred significantly later than in untreated patients [15].

Webb et al. showed that angiotensin II type 1 receptor blocker (ARB), losartan, was also effective as ACEI in reducing proteinuria and safe in children with AS [14]. Nowadays, the use of mineralocorticoid receptor antagonists (MRA) attenuating glomerulosclerosis, interstitial fibrosis, and podocyte injuries come into account. A nonsteroidal selective MRA, finerenone, was found to have an antiproteinuric effect in patients treated with ACEI or angiotensin II receptor blockers. Finerenone reduces renoprotection in patients with type 2 diabetes and chronic kidney disease [16]. Recently, better renal prognosis was observed in patients with different causes of renal insufficiency treated with MRA [17]. A mouse model of Alport syndrome was treated with ramipril, and at week 7, spironolactone was added [18]. Adding spironolactone improved kidney function and reduced proteinuria and fibrosis. A study with MRA in AS patients could be considered. Nowadays, the FIONA OLE study is recruiting children with chronic kidney disease and proteinuria who are treated with ACEI or ARB to study the effect and safety of adding finerenone.

There is no curative treatment of AS; all patients with ARAS, males XLAS older than 2 years, should be treated with ACEI at the time of diagnosis according to current recommendations. Patients with ADAS and females with XLAS should be treated with ACEI after the onset of microalbuminuria [19]. Hypertension has been present since adolescence in some cases, more antihypertensive drugs can be added to gain optimal blood pressure.

## 6. Sodium-Glucose Cotransporter-2 Inhibitors (SGLT2) and Metformin

SGLT2 inhibitors were first shown to slow down the progression of diabetic kidney disease and to lower cardiovascular morbidity and mortality. As SGLT2 inhibitors do not induce hypoglycemia in non-diabetic patients, they were supposed to be safe in other kidney diseases. The nephroprotective effect is caused by vasomodulation of inappropriately dilated afferent arteriole in glomeruli associated with a reduction of intraglomerular pressure followed by a decrease in albuminuria [20]. The DAPA-CKD study with 4304 recruited diabetic and non-diabetic patients, and six patients with AS were also included [21]. Dapagliflozin was well tolerated and decreased the risk of any component of the composite renal endpoint (≥50 eGFR decline, the onset of ESRD, or renal or cardiovascular death) to 44%. A small pilot study with of dapagliflozin in five pediatric patients with AS resulted in a 22% reduction in proteinuria after 12 weeks of treatment [22]. A half-dose dapagliflozin was found to reduce proteinuria in three patients with ARAS and decreased renal function [23]. SGLT2 inhibitors seem to be promising drugs in AS as in other kidney diseases, but more clinical trials, mostly in children, are essential. There are nowadays available in most European countries for patients with stage 3 of chronic kidney disease and proteinuria more than 0.2 g/g creatinine, so SGLT2 inhibitors can be used in patients with Alport syndrome.

Metformin is a biguanide drug that is used in type 2 diabetes. Metformin was shown to slow renal inflammation and fibrosis. The administration of metformin or losartan slowed the progression of renal insufficiency and prolonged the survival in mice with AS [24]. Metformin comes into account because of its low cost and availability. As far as non-diabetic patients are concerned, metformin was found to be a safe drug in patients with autosomal dominant polycystic kidney disease with an estimated glomerular filtration rate over 50 mL/min/1.73 m^2^ [25]. It should not be used in patients with chronic kidney disease stage 4 because of the risk of lactic acidosis [26]. A study from China just started studying metformin in children (from 10 to 17 years) with XLAS or ARAS. The participants will receive treatment of metformin in an initial dose of 500 mg/day, within 2 weeks reaching the maximum tolerated dose (maximum 1500 mg/day). Primary outcomes are either a decrease in proteinuria or a steady estimated glomerular filtration rate (eGFR) from baseline to month 12 or 24 under metformin treatment compared to placebo.

## 7. Bardoxolone

Bardoxolone methyl is a semisynthetic triterpenoid that activates nuclear factor erythroid 2-related factor 2 (Nrf2), a transcription factor that modulates the expression of hundreds of genes involved in inflammation, oxidative stress, and cellular energy metabolism [27]. Bardoxolone diminishes inflammation and formation of reactive oxygen species by reducing NFKB (nuclear factor of activated B-cells) [28]. Bardoxolone has been studied in patients with different kidney diseases such as diabetic nephropathy, IgA nephropathy, FSGS, and autosomal dominant polycystic kidney disease (ADPKD). The BEACON study in diabetic patients was stopped because of an increased rate of heart failure [29].

There is a lack of data using bardoxolone in animal models with AS. The CARDINAL phase 3 study was an international, multicentric, double-blind, placebo-controlled trial in which 157 patients with AS (ages 12–70 years) were randomly assigned to bardoxolone (77 patients) or placebo; 62% of patients had XLAS, the mean baseline eGFR was 62.7 mL/min/1.73 m^2^, and the mean UACR was 141 mg/g. Patients randomized to bardoxolone in CARDINAL experienced better preservation of eGFR at week 48 and 100 in comparison with placebo (differences between the groups 9.2 mL/min per 1.73 m^2^ at week 48 and 7.4 mL/min per 1.73 m^2^ at week 100). The mean decrease of eGFR at week 100 was −1.0 mL/min per 1.73 m^2^ in patients on bardoxolone and −8.4 mL/min per 1.73 m^2^ in patients on placebo. The use of bardoxolone was associated with the worsening of albuminuria, which is reversible, and it is supposed to be caused by decreased tubular reabsorption of albumin after increased glomerular filtration. Decreased tubular reabsorption of albumin is accompanied by the downregulation of megalin. The initial increase of UACR was observed in patients with type 2 diabetes treated with bardoxolone (BEACON trial) and attenuated after six months [29]. Finally, bardoxolone resulted in a significant decrease in albuminuria when indexed to eGFR. The most frequent reversible adverse event was the increase in liver transaminases. To conclude, bardoxolone treatment was effective and led to the preservation of renal function in AS patients treated for 2 years. The long-term effect of bardoxolone on increasing glomerular pressure is uncertain [30], and further longer studies with bardoxolone should follow.

## 8. Endothelin Type A Receptor (ETAR) and Angiotensin II Type 1 Receptor (ARB) Inhibitors

The activation of ETAR plays an important role in kidney and inner ear pathology. Sparsentan, a dual ETAR/ARB inhibitor, reduced proteinuria, increased lifespan, and improved auditory abnormalities in AS mice model [31]. The clinical trial EPPIK (NCT05003986) is recruiting pediatric patients with glomerular disease and proteinuria with a protein-to-creatinine ratio (UPCR) of more than 1 g/g, including patients with AS.

Atresantan functions as a selective ETAR inhibitor. This drug decreased the risk of renal events and proteinuria in diabetic patients, and side effects such as edema and anemia were frequent [32]. The clinical trial AFFINITY (NCT04573920) is currently recruiting patients with glomerular diseases and proteinuria. Patients with AS and proteinuria are expected to be also recruited into the study.

## 9. Anti-MicroRNA-21

MicroRNAs (miRNAs) are short non-coding RNAs that can regulate gene expression by degrading messenger RNA or by inhibiting its translation. MicroRNA-21 is upregulated in kidney diseases. Higher microRNA-21 is associated with higher gene expression modulating tissue repair response after an acute or chronic injury that is followed by inflammatory changes and later fibrosis in the tubulointerstitium in the kidney. The antifibrotic effect of anti-miR-21 oligonucleotides was described in *Col4a3^−/−^* Alport mice. Significant additive effects were found for a combination of anti-miR-21 and ACEI therapies on kidney function and survival in Alport mouse models, especially in the fast-progression model [33]. The expression of microRNA-21 was found to be higher in patients with AS than in controls, and higher expression was associated with a more severe clinical course of AS.

The clinical trial HERA (NCT02855268) with Lademirsen started in patients with AS in 2019. The anti-miRNA-21 molecule was administered as a subcutaneous injection every week for 48 weeks. The study was stopped in 2022 because the effect on kidney function had not been found. We can hypothesize that the effect on tubulointertitial fibrosis in complex genetic diseases as AS is insufficient.

## 10. Lipid-Lowering Agents

Renal lipotoxicity contributes to cell dysfunction and apoptosis, followed by the progression of proteinuria [34]. Impairment in reverse cholesterol transport from cells contributes to glomerulosclerosis and tubulointerstitial damage [35]. Accumulation of esterified cholesterol and triglycerides has been recently found in the kidney cortex in animal models with AS. Statins reduced proteinuria in *Col4a3* knock-out mice, but the positive effect was not proved in patients with AS [36]. Excessive cholesterol can be excluded from cells by ATP binding cassette A1 (ABCA1); however, the expression of this transporter was reduced in animal models with AS [37]. Treatment of mice with AS with hydroxypropyl-β-cyclodextrin (HPβCD) reduced cholesterol content in the kidneys of mice with AS and protected them from the development of albuminuria, renal failure, inflammation, and tubulointerstitial fibrosis. The disadvantage of this molecule is parental administration, non-selectivity, and ototoxicity [38].

A study with a small molecule R3R01 (NCT05267262) is currently recruiting patients with AS and steroid-resistant focal segmental glomerulosclerosis. R3R01 molecule increases levels of functional transporter ABCA1, which can exclude excessive cholesterol from cells. This investigational molecule was designed to reduce fat levels in kidney cells, followed by improvement of kidney function. The study will include at least 20 patients with XLAS or ARAS older than 12 years with UPCR ≥ 1 g/g. The treatment with tablets administered orally twice daily will endure 12 weeks with a primary outcome as the percentage change in proteinuria.

## 11. Hydroxychloroquine

Hydroxychloroquine, the antimalarial drug, is used as a treatment of systemic lupus erythematosus because of its ability to inhibit the immune system by inhibiting Toll-like receptors and suppressing cytokine production. Hydroxychloroquine decreased proteinuria in patients with IgA nephropathy only mildly less than glucocorticoids during 6 months of therapy [39]. The reduction of proteinuria in patients with IgA nephropathy treated with RAAS inhibitors was significant (from 1.9 g/day to 0.9 g/den, *p* < 0.002). Treatment with Hydroxychloroquine is definitely safer than therapy with glucocorticoids.

A randomized, controlled trial in China is currently recruiting XLAS and ARAS patients to study the safety, tolerability, and efficacy of Hydroxychloroquine (NCT04937907), in which 50 pediatric patients will be enrolled and treated for 48 weeks. Treatment with ACEI and/or ARB should be stable at least 4 weeks prior to screening. The primary outcome is the reduction of urinary erythrocyte count. Changes in proteinuria and eGFR are secondary outcomes. There is no clear association between hematuria and the progression of Alport syndrome, but time-averaged hematuria was found to be independently associated with the progression of some other diseases, such as IgA nephropathy [40]

## 12. Future Therapy

### 12.1. Stem Cell Transplantation

The rationale of stem cell therapy in AS is based on the fact that stem cells from healthy donors will migrate to renal glomeruli and will differentiate into podocytes producing new functional products of the glomerular basement membrane. Stem cell transplantation restored the expression of collagen IV, followed by a reduction in proteinuria in irradiated *Col4a3* knockout mice [41]. Other studies did not prove the positive effect of stem cells on kidney function in mice with AS [42]. Stem cell transplantation is a risky procedure used in life-threatening diseases. The results are non in animal models with AS, so it can not be nowadays recommended in patients with AS.

### 12.2. Premature Termination Codon Readthrough Therapy

Shorter α3, α4, and α5 chains of collagen IV are missing the NC1 domain because nonsense mutations in responsible genes are not able to form heterotrimers that are essential for GBM function. Moreover, the reduced mRNA stability because of nonsense-mediated mRNA decay further decreases an expression level. Readthrough truncating nonsense mutation agents come into account as a potential therapy for selected patients with AS. They are able to read through the stop codons on the RNA level (Figure 1A). Cell culture experiments from patients with AS and nonsense mutations are carried out to find out if the mutations are susceptible to readthrough therapy [43].

Readthrough agents are nowadays ongoing in the clinical trials phase 2 in patients with cystic fibrosis and cystinosis. A pilot phase 2 study using nonsense readthrough therapy will evaluate the safety and efficacy of subcutaneously administered ELX-02 (eukaryotic ribosomal selective glycoside) in patients with ARAS and XLAS caused by nonsense mutation (Table 2).

### 12.3. Exon Skipping Therapy

Exon skipping therapy with antisense oligonucleotide or chemical compounds can shift the truncating mutation to an in-frame deletion mutation on the RNA level (Figure 1A) that is mostly less deleterious. Antisense oligonucleotide binds to the exonic splicing enhancer region, disturbing the splicing of the exon and leading to the exon skipping. Promising results were found in vitro and in an animal model with AS in which the exon 21 of *COL4A5* gene skipping was associated with a less severe phenotype with later age of renal failure [44].

### 12.4. Chaperones

A chaperone is a small molecule that enables to stabilize an unfolded protein preventing inappropriately folded protein from being destroyed in the endoplasmic reticulum (Figure 1A). Hexamer formation of the α3, α4, and α5 chains contains multiple pores, which can serve as potential cavities for small chaperons affecting the folding and stability of collagen IV. Chaperones seem to be useful in patients with missense mutations that represent about 40% of all mutations in patients with AS [45]. Chaperones are already used as drugs in the therapy of Fabry disease with missense mutations [46].

There was a study on fibroblast cell lines from men with XLAS using the chaperone sodium 4-phenyl butyric acid (PBA) that increased the expression of collagen IV alpha 5 mRNA [47]. PBA was given orally to *Col4a1* mutant mice, and there was a reduced occurrence of intracranial hemorrhage, but kidney and ocular involvement was not improved [48]. Chaperones could lead to the stabilization of triple-helix formation followed by a formation of more stable GBM.

### 12.5. Protein Replacement

Delivery of full-length protein or a miniature version of collagen IV into the glomerulus is a challenging therapeutic approach. Delivery of the full-length laminin-521 protein to the glomerular basement membrane was used as a therapeutic option in a model of Pierson syndrome [49].

### 12.6. Genome Editing

Gene therapy for AS seems to be a promising therapeutical option, but it is limited to early testing. An inducible trans-gene system in a mouse model with AS restored missing collagen IV in GBM [50].

A gene-editing technology is currently being studied (Figure 1A). The clustered regularly interspaced short palindromic repeats (CRISP) and CRISP-associated protein 9 were used to reverse mutations in podocytes isolated from urine with high correction rates from 44% in the *COL4A3* gene to 58% in the *COL4A5* gene [51]. This system contains single-guide RNA (sgRNA) and an endonuclease. The sg RNA guides Cas9 endonuclease to the specific site on double-strand DNA. The break of DNA is repaired by non-homologous end joining, and the correction is supported by the donor DNA template within the area of interest, which acts as a template for the synthesis of de novo DNA. The manipulation with podocytes is difficult, and gene therapy should be used as early as possible to be effective.

The experiments with X-chromosome reactivation are currently ongoing [52]. During the embryological development of females, one X chromosome is randomly inactivated. Reactivating a healthy copy of the *COL4A5* gene on the inactivated chromosome could improve the prognosis in XLAS females in the future.

## 13. Conclusions

There is currently no curative treatment for AS. Therapy with ACEI since childhood improves the prognosis of AS. SGLT2 inhibitors are safe and effective drugs in AS, but further studies are inevitable. Bardoxolone seems to preserve renal function in AS, but longer studies should follow. New drugs targeting advanced kidney impairment, such as tubular cell injury, inflammation, and fibrosis, did not meet the expectation in clinical trials. The latest therapeutic approaches, such as genome and protein-affecting drugs, seem to be promising molecules for AS treatment in the future.

## Figures and Tables

**Figure 1 ijms-24-05522-f001:**
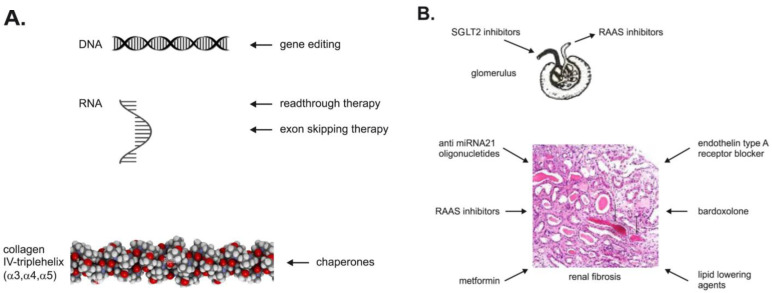
(**A**). Therapeutic targets in AS. Specific therapy is on the level of DNA, RNA, and protein. (**B**). The remaining drugs on the level of glomeruli and tubulointerstitium are unspecific.

**Table 1 ijms-24-05522-t001:** Classification and prognosis of AS [6].

Inheritance	Genetics	Risk to End-Stage Renal Disease
X-linked	mutated *COL4A5* gene	
	males	100%
	females	up to 25%
Autosomal recessive	mutated *COL4A3* or *COL4A4* gene (homozygous or compound heterozygous)	100%
Autosomal dominant	mutated *COL4A3* or *COL4A4* gene (one allele)	0–50%-risk factors (proteinuria, glomerular basement membrane lamellation, focal segmental glomerulosclerosis, hearing loss)
Digenic	mutated *COL4A3* and *COL4A4* genes in the trans position	up to 100%
	mutated *COL4A3* and *COL4A4* genes in the cis position	up to 20%

**Table 2 ijms-24-05522-t002:** Recruiting clinical trials in AS.

Drug	Mechanism of Action	Number of Patients to Be Recruited	Phase and Name of the Study	Primary Outcomes	Secondary Outcomes
**Sparsentan** Alport syndorme and other proteinuric disease	Endothelin A receptor antagonist and angiotensin II type 1 receptor antagonist (ARB)	57	Phase 2 EPPIK	urinary protein-to-creatinine	adverse events
**Atrasentan** Alport syndorme and other proteinuric disease	Endothelin A receptor antagonist	100	Phase 2 AFFINITY	proteinuria	
**Hydroxychloroquine**	Immunomodulatory drug	50	Phase 2 CHXLAS	urinary erythrocyte count	proteinuria
**R3R01** Alport syndrome and focal segmental glomerulosclerosis	Lipid modyfying drug (exlude cholesterol from cells)	50	Phase 2	adverse events urinary protein-to-creatinine ratio	quality of life
**ELX-02**	Ribosomal selective glycoside (premature termination readthrough agent)	8	Phase 2	adverse events	proteinuria collagen IV expression in renal biopsy urinary erythrocyte count
**Finerenone** Proteinuric diseases	mineralocorticoid receptor antagonist (+ACEI or ARB)	219	Phase 3 FIONA	urinary protein-to-creatinine ratio	eGFR, adverse events
**Metformin**	biguanide drug	78	Phase 4	eGFR	safety

## Data Availability

Not applicable.

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
