# Peer review of "Current and Future Therapeutical Options in Alport Syndrome"

_ijms, 2023, doi:10.3390/ijms24065522_

Round 1

Reviewer 1 Report

Reiterova  and Tesar review therapeutical options in Alport syndrome . They add a few items to the previous comprehensive review by E. Chavez et al (Frontiers in Medicine 2022). The authors  add 2 options discussed by the team of B. Hudson (Boudko et al.) in Current Opinion 2022 such as exon skipping therapy and premature termination codon readthrough therapy. Compared to the review by Chavez et al, this one does not mention DDR1 inhibitors and osteopontin blocking agents.

The current paper is of potential interest but suffers , as it is, from too many errors, inaccuracies, typos, inconsistencies etc…. There is thus clearly the need for a very major revision before a revised version could hypothetically be considerer for publication. The present version is very far from reaching that level. The main problems in the submitted version are detailed below , section by section .

 Introduction: The authors use the term “mutations “, whereas the most recent nomenclature suggests to replace it by pathogenic variants. Please detail the abbreviations when first used (ex : GBM). Please specify that in Alport syndrome due to the pathogenic variants, the embryological components (alpha1alpha1alpha2)  persistent and leads to a less functional GBM.

 Prevalence and genetic background: Table 1: Please link this table to  reference 3 (Kashtan et al. K int 2018) and avoid abbreviations (ESRD, GBM, FSGS). Please use the term of pathogenic variants. The prevalence of XLAS is no more considered as 80 % nowadays due to the higher prevalence of ADAS. Please clarify this part. “All patients with ARAS and XLAS males should be treated with RAS inhibitors” please add the reference pf the new guidelines here (ref 16) and add “since diagnosis.”

 Clinical course and progression: the prognosis depends on  the type of mutation in XLAS men but also more generally and critically on the mode of inheritance. Genotype-phenotype correlations are well established for XLAS men but not for others (ARAS, ADAS, XLAS women). The paragraph on digenic Alport syndrome is already detailed in prevalence and genetic background, please remove it here.

TREATMENT:  The statement “ young patients with comorbidities” refers to which age and which comorbidities? Please clarify.

Figure 1. Part A is not useful and quite simplistic (?). Please add more detail schemes with the mechanisms of action of gene editing, of readthrough therapy and exon skipping therapy as well as chaperones (typo mistakes : chaperonEs, please correct also in the manuscript).

Table 2. This table is not clear and contains several mistakes. The title needs to be changed as all the studies mentioned are recruiting (cfr infra). Typo mistake in Sparsentan. Sparsentan is also an AT1R inhibitor, please add. Please add in the table the name of the study if available (for example EPPIK, aFFINITY…) R3R01 please detail the mechanism of actions of this agent in the table and in the manuscript. This study is currently recruiting as you mention in the manuscript, please correct it in the table. ELX-02 is also currently recruiting. Please correct. The “intervention column” is not useful. A single column with the molecule and its action may be clearer. Number of patients of what? To be recruited? The last column is useless and has no title, what is the purpose of this column? Add a column with the phase of the study and the type (open-label, controlled etc…), a column with the study outcomes (primary and secondary) and also the inclusion criteria (type of Alport syndrome, ages…). The study with hydroxychloroquine is  not only for XLAS patient. Its primary outcome is unusual as it is the reduction of urinary erythrocyte count which may need a special discussion in the manuscript.

 RAAS inhibition: Please add the reference of Rubel et al. Organoprotective effect of Spironolactone on top on Ramipril therapy in mouse model for Alport syndrome. J Clin Med 2021. FIONA OLE study is recruiting children to study Finerenone in CKD proteinuric children. Please add it in the table and in the manuscript.

The optimal blood pressure does not require a mention (percentile goal is only for children).

 SGLT2-i and Metformin : Please clarify  this sentence  “as SGLT2 inhibitors do not induce hypoglycemia in non-diabetic patient  they were supposed to be nephroprotective in other kidney diseases” or remove it.  Six AS patients were included in DAPA-CKD. Please specify the composite renal endpoint of DAPA-CKD. You may add other small studies in AS (Bockenhaus, Gross in Cells 2021; Song et l. K Int Reports 2022; Song et al. Dapagliflozin in ARAS. K Int Reports 2022).

A study from China just started studying Metformin in AS in children, please detail and add in your table.

 Bardoxolone : Please detail the worsening of albuminuria in CARDINAL study and the reversibility (after washout?). This point is a big concern for a proteinuric nephropathy such as AS.Worsening of albuminuria is supposed to be caused by decreased tubular resorption, please add “supposed”.

 Endothelin receptor antagonist: typo mistakes : eodema and anemie (edema or oedema and anemia)

Anti-microRNA-21: Please clarify the sentence “ MicroRNA-21 is upregulated in kidney diseases regulating the repair response…”

 Lipid lowering agents : please detail the mechanisms of R3R01.

 Hydroxychloroquine:  please detail the study with IgA nephropathy and comment. As mentioned above, the study in China is also open for ARAS and the primary endpoint is the urinary erythrocyte count, please correct, detail and comment.

Future therapy

 Stem cell transplantation : please detail the rationale of its use

Premature termination codon readthrough therapy : please make a figure with this specific mechanism. Please change the sentence “derived for aminoglycoside derivatives” and clarify the mechanism. Please change the reference used (36, Omachi et al. to Omachi et al. Nanolu reporters…. iScience 2022). Please change the sentence “ Genetic therapy using nonsense readthrough therapy is prepared for a clinical trial” and remove the details about the study, just refer it to the Table 2 and adapt the Table as suggested above.

Exon skipping therapy: please refer to a figure

Chaperones: please correct the typo chaperone. Missense mutations represent (and not cause) about 40 %, please add a reference for this statement. Please clarify the previous study using chaperones (which cells?)

Protein replacement : correct the typo : glomerular basement membrane and not membrany

 Genome editing : remove the artificial chromosome transgenic line, not useful. X chromosome reactivation is mentioned by Chavez et al.  Please cite this reference or remove the section  or give examples of other diseases where this is applied or studied.

Conclusion : please add a comment about proteinuria and bardoxolone

References : please unify references with “et al.” after 3 authors. There are typo mistakes in titles of articles (capital letter missing in England reference 1, in Alport syndrome reference 8, accent on “system” reference 8, “recptor” reference 15 …).

Reference 11 is the same as reference 13. Reference 10 is incorrect, the authors probably refer to Gross O et al. Kidney International 2012.

Author Response

Reply to the reviewer 1

  1. Mutations were replaced by pathogenic variants. We specified that embryological components stay persistent.
  2. In Table 1 , we avoided abbreviations, refence 6 was added.
  3. 19 – new guidelines are mentioned at the end of pararagraph with RAAS inhibition.
  4. We removed the paragraph on digenic Alport from clinical course.
  5. Young patients (children and adults according to study entry criterias) with AS without severe comorbidities such as diabetes mellitus, cardiovascular disease, tumours are eligible candidates for clinical trials.
  6. 1- A. The mechanims are explained in text, this simple scheme shows only different level of DNA, RNA,protein, where the therapeutical options come into account. Chaperones were corrected.
  7. I corrected table 2, I added name of recruiting studies and outcomes.
  8. RAAS inhibition- I added – Rubel et al. Study and I mentioned FIONA OLE study.
  9. I corrected nephroprotective to safe (in SGLT2 inhibitors), I defined composite end point(≥50 eGFR decline, the onset of ESRD or renal or cardiovascular death) and I added small Song et al study. Metformin study in China is added to manuscript and  table 2.
  10. I discussed the effect of bardoxolone on proteinuria in more details.

The use of bardoxolone was associated with worsening of albuminuria which is reversible and it is supposed to be caused by decreased tubular reabsorption of albumin after increased glomerular filtration. Decreased tubular reabsorption of albumin is accompanied by downregulation of megalin. The initial increase of UACR was observed in patients with type 2 diabetes treated with bardoxolone (BEACON trial) and attenuated after six months [29]. Finally, bardoxolone resulted in a significant decrease in albuminuria when indexed to eGFR.

  1. I corrected edema and anemia.
  2. I clarified anti-micro RNA

Higher microRNA-21 is associated with higher gene expression modulating tissue repair response after acute or chronic injury that is followed by inflammmatory changes and later fibrosis in tubulointerstitium in kidney.

  1. Mechanism of R3R01

R3R01 molecule increases levels of functional transporter ABCA1 which can exclude excessive cholesterol from cells. This investigational molecule was designed to reduce fat levels in kidney cells followed by improvement of kidney function.

  1. Hydroxychloroquine- I put more information

      Hydroxychloroquine decreased proteinuria in patients with IgA nephropathy only mildly    less than glucocorticoids during 6 months therapy [39]. The reduction of proteinuria in patients with IgA nephropathy treated with RAAS inhibitors was significant (from 1.9 g/day to 0.9 g/den, P 0.002). Treatment with hydroxychloroquine is definitely safer than therapy with glucocorticoids.

Randomized, controlled trial in China is currently recruiting XLAS and ARAS patients for studying the safety, tolerability and efficacy of Hydroxychloroquine (NCT04937907). 50 pediatric patients will be enrolled and treated for 48 weeks. Treatment with ACEI and/or ARB should be stable at least 4 weeks prior to screening. Primary outcome is the reduction of urinary erythrocyte count. Change in proteinuria and eGFR are secondary outcomes.  There is not clear association between hematuria and progression of Alport syndrome, but time-averaged hematuria was found to be independently associated with the progression of some other diseases such as IgA nephropathy [40]

  1. Stem cell transplantation – rationale

The rationale of stem cell therapy in AS is based on the fact that stem cells from healthy donors will migrate to renal glomeruli and will differentiate into podycyte producing new functional products of glomerular basement membrane. 

  1. I removed derived for aminoglycoside and nonsense readthrough therapy is prepared for clinical trial, I corrected the reference.
  2. I referred to Fig., but the mechanisms are explained in manuscript.
  3. I corrected chaperones, I added the ref. to 40% of missense mutations, I added fibroblast cell lines.
  4. I corrected membrane.
  5. I removed arteficial chromosome.

I unified references.

Reviewer 2 Report

Review of ijms-2170594

Current and future therapeutical options in Alport syndrome

Reiterová J and Tesař V1

General comments

This manuscript describes the updated therapeutical options in Alport syndrome. This is a well written and useful contribution. I have annotated the manuscript with several minor corrections, which I believe will improve the readability of the paper.

Specific comments

1, In abstract, what the definition of “young patients”? Do the authors indicate children?

2, In abstract, abbreviation “DAPA-CKD” should be defined.

3, In introduction, there was no reference. Could the authors add appropriate references?

4, In figure1A, what is the meaning of triple helix (23, 24, 25)? Dose the number have any meanings?

5, There appears to be an arrow in the image of renal fibrosis in Figure 1B. If it has meaning, please add the explanation. The arrows of RAAS inhibitors appear in the opposite direction.

Author Response

Reply to the reviewer 1

  1. I corrected young patients to young adults in abstract.
  2. I defined abbreviation DAPA CKD in abstract.
  3. I added references in Introduction.
  4. I cut out the number under triple helix, it was transmission errror.
  5. The arrow in the figure is not related to arrows with therapeutical options.

Reviewer 3 Report

The authors proposed a review for the treatment of Alport syndrome.

The manuscript is well-written, and informative.

However, I have some comments on the manuscript.

First, the diagnosis of Alport syndrome (AS)is sometimes challenging.

Even though patients undergo renal biopsy, AS can be overlooked without a suitable family history or electron microscopy. Detecting gene mutations are sometimes difficult even in patients with historical findings specific to AS, such as collagen type IV staining, and electron microscopy findings. This might be because the mutations can exit the intron area.

I understand that this review is focused on the treatment of AS, but the authors should describe that diagnosis of AS can be challenging. On the other hand, there will be several treatment strategies for AS in the future, which means that accurate diagnosis is mandatory to adopt these strategies, especially in AS-specific treatment.

Second, I do not agree with showing metformin as a candidate drug. Differed from other drugs, this drug is not under clinical trial on humans. Additionally, the cited paper did not administer metformin with losartan in mice, but metformin “or” losartan. Although the authors described that this drug should not be used in CKD stage 4 and over, I do not agree with citing only animal-based evidence. In fact, the evidence of metformin for non-diabetic patients is only limited.

Minor points

Page 1, The authors should cite some papers on “Despite the fact that heterozygous females usually have much less severe clinical course, there are nowadays recommended to be classified as having Alport syndrome instead of being carriers of Alport syndrome”

Page 2, “SRD” would be ESRD. 

Author Response

Reply to the reviewer 2

  1. I added at page 4 the paragraph about significance and possible pitfalls of an accurate diagnosis of AS for future treatment.

A heterogeneous clinical course needs the co-operation of nephrologists, pathologists and clinical and molecular genetics to find out an accurate diagnosis than is mandatory for therapeutical strategies. A detailed family history is helpful from nephrologists, the diagnosis can be confirmed from electron microscopy after renal biopsy by pathologist. The genetic testing can identify a pathogenic gene variant leading to an individualized treatment in future. On the other hand, we have to take into account that the mutation is not even with new generation sequencing identified in about 10% of patients with AS.

  1. I added citations 22,23 about metformin treatment and risks in nondiabetic population
  2. I put a citation 3 related to heterozygous females with AS. SRD was corrected ESRD.

Round 2

Reviewer 1 Report

The paper is undoubtedly imporved, but there are still a number of issues requiring attention. Below a listing, not necessarily comprehensive. It is the job of the authors  (including the senior one) to read carefully their paper before resubmission

1) The result from renal biopsy does not help us with a prognosis, the risk of AS progression ADD again here IS DETERMINED  by the type of mutation in XLAS men but also on the mode of affected COL4 genesinheritance. Genotype-phenotype correlations are welL estab-lished for XLAS men but not for others forms.

2) stem cells section : The results are non in animal models with AS WHAT DO THE AUTHORS MEAN?

3) The figure is of moderate interest at best, could be improved (see previous round of review)

4) in the table of ongoing studies, the DRUG column should aslo include in title the target population

5).....

Author Response

Reply to the reviewer 1

The result from renal biopsy does not help us with a prognosis, the risk of AS, the risk of AS  progression is determined by the type of mutation in XLAS men but also depends on the mode of affected COL4 genes inheritance. Genotype-phenotype correlations are well established for XLAS men but not for other forms.

              The results are ambiguous in animal models with AS, so it can not be nowadays                   recommended in patients with AS.

                Column- drug, target population added
